# The Impact of Starting Positions and Breathing Rhythms on Cardiopulmonary Stress and Post-Exercise Oxygen Consumption after High-Intensity Metabolic Training: A Randomized Crossover Prospective Study

**DOI:** 10.3390/healthcare12181889

**Published:** 2024-09-20

**Authors:** Yuanyuan Li, Jiarong Wang, Yuanning Li, Dandan Li, Yining Xu, Yi Li

**Affiliations:** 1Physical Education Department, Shandong Pharmaceutical and Food Vocational College, Weihai 264210, China; lyy5996@126.com; 2Faculty of Educational Studies, University Putra Malaysia, Selangor 43400, Malaysia; jiarongwang0719@gmail.com; 3Faculty of Sports Science, Yanshan University, Qinhuangdao 066000, China; 17394672612@163.com; 4College of Physical Education and Sports, Beijing Normal University, Beijing 100084, China; 17861520032@163.com; 5Faculty of Sports Science, Ningbo University, Ningbo 315211, China; 6China Volleyball College, Beijing Sport University, Beijing 100084, China

**Keywords:** metabolic, cardiopulmonary, prospective, cox regression, random survival forest

## Abstract

*Background:* The exploration of optimizing cardiopulmonary function and athletic performance through high-intensity metabolic exercises (HIMEs) is paramount in sports science. Despite the acknowledged efficacy of HIMEs in enhancing cardiopulmonary endurance, the high metabolic stress imposed on the cardiopulmonary system, especially for amateurs, necessitates a scaled approach to training. *Objective*: The aim of this study is to ascertain whether adjustments in the initiation posture and the adoption of an appropriate breathing strategy can effectively mitigate the cardiopulmonary stress induced by HIMEs without compromising training efficacy. *Methods:* Twenty-two subjects were recruited into this study. The post-exercise heart rate (PHR) and post-exercise oxygen consumption rate (POCR) were collected within 30 min after exercise. A two-way ANOVA, multi-variable Cox regression, and random survival forest machine learning algorithm were used to conduct the statistical analysis. *Results:* Under free breathing, only the maximum POCR differed significantly between standing and prone positions, with prone positions showing higher stress (mean difference = 3.15, *p* < 0.001). In contrast, the regulated breathing rhythm enhanced performance outcomes compared to free breathing regardless of the starting position. Specifically, exercises initiated from prone positions under regulated breathing recorded a significantly higher maximum and average PHR than those from standing positions (maximum PHR: mean difference = 13.40, *p* < 0.001; average PHR: mean difference = 6.45, *p* < 0.001). The multi-variable Cox regression highlighted the starting position as a critical factor influencing the PHR and breathing rhythm as a significant factor for the POCR, with respective variable importances confirmed by the random survival forest analysis. These results underscore the importance of controlled breathing and starting positions in optimizing HIME outcomes. *Conclusions:* Regulated breathing in high-intensity exercises enhances performance and physiological functions, emphasizing the importance of breathing rhythm over starting position. Effective training should balance exercise volume and technique to optimize performance and minimize stress, reducing overtraining and injury risks.

## 1. Introduction

In the field of sports science, the exploration of better approaches to optimize cardiopulmonary function and athletic performance through diverse training methodologies has always been a focal point of research [1]. High-intensity metabolic exercises (HIMEs), recognized for their ability to significantly elevate heart rate within a brief interval, are widely acknowledged as an efficacious strategy for enhancing cardiopulmonary endurance [2,3]. Moreover, the versatility of HIMEs, encompassing various levels of progression and regression, allows for both amateurs and elite trainers to tailor their training according to individual requirements [4]. This adaptability has solidified the presence of HIMEs in the training plans of both amateur trainers and elite athletes alike [5].

However, the high metabolic stress inherent in HIMEs imposes significant stress on the cardiopulmonary system, particularly for amateurs who often find the rapid escalation in heart rate intolerable, leading to an apprehension towards HIMEs and affecting adherence and outcomes [6,7]. This issue stems from the dynamic shifts in the body’s center of gravity during HIMEs, such as jumping jacks, burpees, and air squats, necessitating the heart to rapidly adjust to the fluctuating demands for blood return and output [8,9]. Throughout these exercises, as the body swiftly transitions from higher to lower positions and back, the redistribution of blood necessitates an accelerated cardiac pumping rate to maintain adequate circulation and oxygen supply throughout the body. Consequently, this rapid increase in heart rate exerts substantial pressure on the cardiopulmonary system [10].

Currently, trainers frequently adopt a scaled approach to reduce the complexity and intensity of HIMEs, ensuring the quality and efficacy of training [11]. For example, modifications such as substituting the jump in standard burpees with steps or omitting the push-up phase to simplify the exercise have positively impacted trainers’ compliance and stickiness as well as the efficacy. However, this inevitably leads to a reduction in training volume as participants progress towards standard movements. Moreover, altering the fundamental structure of exercise movements may compromise the specificity of transfer, causing individuals accustomed to the scaled versions to experience significant discomfort upon transitioning to standard movements [12,13].

Furthermore, the role of breathing rhythm in modulating cardiopulmonary stress during exercise, particularly within the context of high-intensity metabolic exercises (HIMEs), is critical. Proper breathing control not only aids in maintaining heart rate within a manageable range but also optimizes oxygen uptake and carbon dioxide expulsion, enhancing recovery rates post-exercise [14,15]. Studies have shown that by regulating breathing patterns, athletes can more effectively reduce the cardiopulmonary stress induced by exercise, expedite blood circulation, and thereby shorten the recovery period, increasing the overall training efficacy [16,17,18]. In cardiopulmonary function, rhythmic deep breathing increases pulmonary ventilation and improves cardiac pump efficiency. This enhanced cardiopulmonary function means the heart does not need to work as hard to maintain the same oxygen supply [19,20]. Hence, controlled breathing not only alleviates the burden on the heart but also helps rapidly return the heart rate to resting levels post-exercise, speeding up recovery from high-intensity activities.

Some previous studies have increasingly focused on optimizing cardiopulmonary function through various exercise protocols, with particular attention to the physiological impacts of breathing patterns and body positions during exercise. For example, a study published in 2016 by Forton’s team demonstrated that body position does not influence the exercise stress echocardiography of the pulmonary circulation or aerobic capacity as gauged by maximum oxygen uptake. However, the semi-recumbent position leads to a lower maximum workload, with a broad variation in the predicted cardiac output across different workloads or oxygen uptake levels. Thus, describing pulmonary vascular function is more accurately conducted by relating vascular pressures to cardiac output rather than workload or oxygen uptake [21]. Another study conducted in 2017 evaluated cardiopulmonary function in different body positions during recovery after maximal exercise with fifteen male university students. It found that the trunk forward leaning position significantly enhanced pulmonary ventilation post-exercise by reducing minute ventilation volume and oxygen uptake compared to sitting or supine positions [22]. Moreover, in 2016, a study assessed the effects of different body positions on heart rate, pulmonary, and muscular oxygen uptake kinetics during leg exercises in ten healthy individuals. It found significant differences in pulmonary oxygen uptake kinetics across positions, suggesting that venous return and cardiac output distortions must be considered for an accurate analysis of exercise responses at the muscular and cellular levels [23]. These studies, however, predominantly examined these effects in isolation without integrating the variable of body position, which can be crucial in understanding the full impact on cardiopulmonary dynamics.

This study specifically focuses on the effects of regulated versus free breathing patterns on heart rate and post-exercise oxygen consumption (POC) among experienced trainers. We hypothesize that different body positions during exercises—prone and standing—impact the cardiopulmonary stress experienced. In the prone position, the alignment reduces gravitational stress on heart function, facilitating enhanced blood and oxygen delivery to extremities, which aids in recovery and endurance performance. Conversely, in the standing position, the heart faces greater demand due to the vertical direction of blood pumping. The aim of this study is to ascertain whether adjustments in the initiation posture and the adoption of an appropriate breathing strategy can effectively mitigate the cardiopulmonary stress induced by HIMEs without compromising training efficacy. The burpee exercise, initiated from a prone (from chest to ground) and standing position (from standing position), is selected as the subject of this study.

## 2. Methods

### 2.1. Study Design

This study was a randomized crossover study conducted in Faculty of Sports Science, Ningbo University, Ningbo, China. The study recruited a group of college students who exercise regularly to participate from 15 May 2024, to 15 June 2024. The institutional review board sanctioned this study prior to the initiation of recruitment and the commencement of data gathering. Prior to their engagement, all participants were thoroughly briefed on the study’s objectives, methodology, potential hazards, and precautionary measures; this study received the endorsement of the Ethics Committee of Beijing Sport University, bearing the ethical registration identifier BSU20240008.

### 2.2. Participants

#### 2.2.1. Inclusion Criteria

The inclusion criteria of participants were as follows: (1) from 18 to 22 years old; (2) free from endocrine, metabolic, neuromuscular, and musculoskeletal disorders; (3) with Body Mass Index (BMI) from 18.5 kg/m^2^ to 23.9 kg/m^2^ [21]; (4) without any diseases that are not clinically recommended for physical activity; (5) able to sustain engagement in physical activities of moderate-to-high intensity, which correspond to physical activities with metabolic equivalents (METs) larger than 3, no less than thrice weekly [22].

#### 2.2.2. Exclusion Criteria

The exclusion criteria of participants were as follows: (1) were under 18 years old or over 22 years old; (2) had endocrine, metabolic, neuromuscular, or musculoskeletal disorders; (3) had BMI below 18.5 kg/m^2^ or above 23.9 kg/m^2^ [21]; (4) were clinically required not to participate in any physical exercise; (5) had lack of engagement in physical activities or the intensity of physical activity was lower than 3 METs.

### 2.3. Interventions

The study selected “Burpee” as the exercise movement. Participants were mandated to execute the burpee exercise protocol across 4 distinct testing days, adhering to a randomized sequence. Each testing day was structured into 3 parts: warm-up, testing, and cool-down. (1) For the warm-up, participants were commenced with 5 min of dynamic stretching and activation under the guidance of a professional trainer, followed by a 5-min medium-intensity session on a power air-bike (Assault Air Bike, Assault Fitness Products., Ste. B Carlsbad, CA, USA) calibrated to 5 to 6 points on the Borg Rating of Perceived Exertion (RPE), whose range of scores was 0 to 10. (2) In the testing phase, participants were compelled to fulfill the burpee exercise protocol in accordance with the sequence randomly allocated to them. (3) During the cool-down, under the assistance of a professional trainer, participants engaged in a 5-min treadmill walk at a self-determined comfortable pace, succeeded by 10 to 15 min of stretching and manual relaxation administered by the trainer.

To investigate the hypothesis, specific breathing patterns were assigned to each exercise position—prone and standing. The rationale for selecting these patterns was based on their potential to optimize physiological responses and enhance the overall effectiveness of the training protocol under different gravitational stresses: (1) In the prone position (burpee from chest to ground), participants were instructed to employ deep inhalations when lifting themselves from the ground. This protocol was intended to maximize abdominal expansion and diaphragmatic movement, thereby facilitating venous return and enhancing cardiac filling during upward movement. This position was hypothesized to reduce the heart’s workload, allowing for more efficient blood flow and reduced cardiopulmonary stress. (2) In the standing position (burpee while standing), during the transition back to the ground, controlled exhalations were emphasized. This protocol helped in stabilizing the core and increasing intra-abdominal pressure, which supported the spine during high-impact movements and ensured efficient expulsion of carbon dioxide built up during intense exertion. The heart continued to pump blood vertically, which might create greater cardiopulmonary stress compared to the prone position.

Therefore, there were 4 burpee exercise protocols assigned in random order to each participant: (1) regulated breathing rhythm + start from standing position, known as RS; (2) regulated breathing rhythm + start from prone position, known as RP; (3) free breathing rhythm + start from standing position, known as FS; and (4) free breathing rhythm + start from prone position, known as FP. The constituent steps and phase-specific requisites of the burpee movement were illustrated in Figure 1.

During the burpee exercise, participants must execute 20 burpees every minute (referred to as Every Minute on the Minute, EMOM), with no allowances for exceeding 20 repetitions [23]. Early completion permits a resting period until the onset of the next minute; otherwise, it is a failure to complete 20 burpees in 1 min.

While participants may determine their movement’s pace and rhythm, both the starting position and breathing pattern must align with the specific directives of that day’s protocol. Concerning breathing rhythm, a “Free” approach allows participants to independently select their manner and tempo of respiration. Conversely, employing a “rhythmic” breathing strategy necessitates that participants adjust their moments of respiration to correspond with their initial posture: (1) within the “standing position”, participants are permitted a singular breath following the concluding leap of a burpee and prior to the commencement of the subsequent burpee; (2) within the “prone position”, a solitary breath is sanctioned during the initial phase of the burpee’s push-up motion, specifically when the chest makes contact with the ground, preceding the push-up action. Finally, for the “standing position”, participants are required to noticeably pause while in the standing posture, yet they must not linger on the ground during the push-up segment; the chest’s contact with the ground should adhere to a “touch and go” principle. In contrast, within the “prone position”, a palpable pause is mandated when the chest contacts the ground, whereas prolonged standing during the standing phase is prohibited; the transition between standing and squatting necessitates an “up and down” motion. The requirements of breathing, positions, and conditions for stop test in each group were listed in Table 1.

FitMate^®^ EMD gas metabolism analyzer (Version K5, COSMED Co., Ltd., Rome, Italy) was used to assess the oxygen consumption of each participant during the whole test and ensure all participants could fulfill the requirements of the tests. The Version K5 FitMate^®^ EMD gas metabolism analyzer features a portable and wearable design. It not only measures the participant’s oxygen consumption during the test but also monitors adherence to the breathing rhythm protocol. During the test, a researcher continuously monitored the breathing pattern of the participants by observing the ventilatory parameters displayed on the screen of the gas metabolism analyzer. If a participant took a breath outside the designated time of breathing, the screen immediately showed a change in ventilatory values. In such cases, the test was deemed a failure and promptly terminated, and the participant was asked to take a full rest. The test was rescheduled and conducted again after a 48 h rest period.

Based on the specified requirements for breathing, positions, and conditions for stopping the test in each group, under the regulated breathing rhythm protocol, if the participant breathed at unauthorized times, the airflow meter on the FitMate^®^ EMD mask detected a change. This affected the oxygen consumption readings displayed on the monitor, necessitating the termination of the test for the participant.

The whole experiment process is shown in Figure 2. Within the testing day (28 February to 28 May 2024), participants were afforded the discretion to schedule their testing days yet were mandated to (1) refrain from engaging in vigorous physical activities (METs > 6) 48 h prior to a testing day; (2) abstain from alcohol consumption and the intake of contraindicated medications for exercise 72 h before; (3) avoid consumption of substance that could influence the cardiovascular system, such as caffeine and taurine, 24 h in advance; (4) ensure a minimum of 8 h of sleep the night before a testing day; and (5) ensure that they did not experience any respiratory-related illnesses throughout the duration of the testing.

### 2.4. Outcomes

Upon enrollment in the study, participants were required to furnish fundamental details such as sex, age, body height, bodyweight, training experience, and preferences regarding training activities. Prior to commencing the testing procedures on each testing day, every participant was instructed to assume a supine position, eyes closed, and remain stationary for 30 min to facilitate the measurement of their resting oxygen consumption rate and resting heart rate. During the burpee exercise, the total repetitions would be recorded. After the burpee exercise, the average heart rate and post-exercise oxygen consumption rate of participants were collected immediately after and at the 5th, 10th, 15th, 20th, 25th, and 30th minute to evaluate the cardiopulmonary stress and exercise overall intensity.

#### 2.4.1. Post-Exercise Heart Rate (PHR)

The collection of participants’ post-exercise heart rates (PHRs) was recorded via a Polar^®^ Heart Rate Sensor (Version H10, Polar Research Library, Tampere, Finland) [24]. The PHR metric served to quantify the cardiopulmonary stress exerted on individuals by various burpee exercise protocols, thereby gauging the intensity level of each protocol. A sustained elevation in heart rate following a training session could indicate a higher level of training intensity. The variance in heart rate alterations prompted by distinct training regimens facilitates the evaluation and comparison of the intensity levels of these programs. Conversely, a gradual decrement in heart rate post-training may signify that the training intensity was excessively high [25].

#### 2.4.2. Post-Exercise Oxygen Consumption Rate (POCR)

FitMate^®^ EMD gas metabolism analyzer (Version K5, COSMED Co., Ltd., Rome, Italy) was used to collect the post-exercise oxygen consumption rate (POCR) of each participant. Calibration of this sophisticated device necessitated the input of specific environmental conditions, including the temperature, atmospheric pressure, and relative humidity of the indoor surroundings [26]. Preceding the formal evaluation, participants were required to recline in a state of rest for 30 min. Following this period, they were instructed to don a facial mask, engineered to encapsulate the nose and mouth fully, facilitating the capture of exhaled breath. This captured breath was then conveyed to the gas metabolic analyzer for a duration of 5 min, with the objective of determining the mean oxygen consumption within this timeframe, which served as the baseline oxygen consumption for that testing day.

Subsequently, participants disengaged the mask and proceeded to complete the day’s scheduled burpee exercise in a predetermined sequence. Immediately upon the conclusion of these exercises, the mask was re-applied for an additional 30 min assessment of oxygen consumption rate. Data pertaining to oxygen uptake were systematically gathered at the 1st, 5th, 10th, 15th, 20th, 25th, and 30th minute post-exercise, facilitating the computation of an average oxygen consumption rate for each enumerated interval.

POCR could offer critical insights into physiological and metabolic functions. Following high-intensity exercise, an elevation in the rate of oxygen consumption was observed, indicating an increase in energy expenditure by the body to return to a state of rest. This encompassed the energy required for replenishing muscle phosphocreatine (PC) stores, eliminating lactic acid from the bloodstream, and facilitating cellular repair. The variances in POCR following diverse training regimens served as a valuable metric for evaluating the intensity and efficacy of the exercises. Typically, training sessions of greater intensity or difficulty resulted in an elevated POCR, reflecting the body’s increased demand for energy and extended recovery period [27,28].

### 2.5. Statistical Analysis

The SPSS Software 17.0 (SPSS, Inc., Chicago, IL, USA) was used to conduct statistical analyses. Descriptive statistical analysis and ANOVA were used to conduct the baseline description. The employment of two-way ANOVA served to ascertain the presence of significant disparities in maximum PHR, average PHR, maximum POCR, and average PHR following the four variants of burpee exercise protocols (RS, RP, FS, and FP), setting the statistical significance threshold (α) at 0.05. Post hoc testing, employed for elucidating differences in main and interaction effects among groups, utilized the Bonferroni–Holm correction skill for adjusting *p*-values derived from these tests. Given the four distinct burpee exercise protocols, the statistical significance threshold for post hoc evaluations was accordingly defined as 0.05/6 = 0.008.

Additionally, this study collected PHR and POCR at 7 time points after the burpee exercise. Using repeated measures ANOVA would increase the model’s complexity and reduce the interpretability of the results. Consequently, this study treated the declining trend of PHR and POCR after exercise as a “decay process” and assumed the fall of PHR and POCR to resting levels as an “event”. It used multi-variable Cox regression, with the ratio of PHR and POCR to resting oxygen consumption rate as the dependent variable, to explore the changing trend of PHR and POCR over time and the effects of breathing rhythm and starting position on this trend [29,30].

Lastly, the study used the random survival forest (RSF) machine learning algorithm to verify the impact of breathing rhythm and starting position on the change in PHR and POCR over time [31]. Within the random survival forest (RSF) framework, the procedure initiates with the computation of the absolute values of Survival Shapley Additive Explanations (*|SurvSHAP*(*t*)*|*, also known as Shapley values, quantifying the mean contribution of each feature towards the model’s predictive accuracy across all conceivable feature permutations) at each temporal juncture, as in Equation (1). In the RSF of the PHR and POCR over time, the *SurvSHAP*(*t*) value signified the average contribution of each parameter towards forecasting the PHR or POCR at a particular temporal marker *t*. This metric was derived by either simulating or calculating the contributory impact of various feature subsets on the likelihood prediction of a notable decline in PHR or POCR at the specified time point *t* [32,33].
(1)$$SurvSHAP\left(t\right)=\sum S\subseteq N\{i\}\frac{\left|S\right|!\left(N-\left|S\right|-1\right)!}{N!}\left[v(S⋃\{i\}\right])-v(S)]$$
where *N* was the overall number of the parameters; *S* represented any subset of the collection of features, excluding the feature parameter *I*; *|S|* denoted the quantity of elements comprised within the regulated *S*; *v*(*S*) signified the contribution value of features within the regulated *S* to the model’s performance; *v*(*S*⋃{*i*}) was the contribution value to the model’s predictive accuracy subsequent to incorporating feature *i* into the regulated *S*; and *SurvSHAP*(*t*) delineated the Shapley value attributable to feature *i*.

Subsequently, the feature importance was ascertained according to the aggregated *|SurvSHAP*(*t*)*|*, achieved by amalgamating or summing the absolute values of all *SurvSHAP*(*t*) values across temporal points. This process facilitated a comprehensive assessment of each feature’s overall significance, enabling a comparison of the influence exerted by the breath protocol and starting position on the temporal evolution of PHR and POCR [34].

The multi-variable Cox regression and RSF analysis were conducted in R studio software (Version 402.pro1, RStudio, PBC, Boston, MA, USA).

## 3. Results

Eventually, 30 students from Ningbo University were recruited, and 22 participants completed the test. The anthropometric information of participants and the baseline analysis of heart rate and oxygen consumption before the test were shown in Table 2.

The results of participants’ performance and the descriptive statistical analysis in different burpee exercise protocols were listed in Table 3. According to Table 3, all outcomes, the overall repetitions, maximum PHR, average PHR, maximum POCR, and average POCR, had statistically significant differences between groups (*p* < 0.05). Moreover, the results of post hoc tests after Bonferroni–Holm correction (the right side of Table 3, which was the league table of post hoc test, where every cell delineated the mean difference and standard error between the group denoted by the row in comparison to the group represented by the column) showed that under the “free breath rhythm” condition between burpee exercises starting from the standing position and those starting from the prone position, only the maximum POCR indicator showed a significant difference (the latter being higher than the former, mean difference = 3.15, *p* < 0.001). Nevertheless, within the “regulated breath rhythm” condition, with the exception of the maximum PHR, all other measured outcomes significantly surpassed those observed under the “free breath rhythm” condition irrespective of the exercise commencing from a standing or prone starting position. Furthermore, within the regime of “regulated breath rhythm”, distinct statistical variances were discernible solely in the maximum PHR and average PHR metrics when comparing burpee exercises initiated from a standing position to those beginning in a prone position, with both the maximum and average PHR measurements for the RP group significantly exceeding those of the RS group (maximum PHR: mean difference = 13.40, *p* < 0.001; average PHR: mean difference = 6.45, *p* < 0.001).

Figure 3 illustrated the temporal trend of the PHR and POCR, revealing that across all four burpee exercise protocols, both the PHR and POCR exhibited a downward trajectory, with remarkably consistent trends. Notably, the rate of decline within the initial 5 min post-exercise was significantly greater than the decline observed between 5 and 30 min post-exercise.

Table 4, as well as the forest plots in Figure 4, showed the results of the multi-variable Cox regression modeling of the PHR and POCR. According to the results of the multi-variable Cox regression modeling of the PHR and POCR, for the RHR, the hazard ratio of the starting position factor reached 1.574, which is higher than the hazard ratio of the breath protocol factor (0.779). At the same time, the Z-test result of the starting position factor showed statistical significance (*p* = 0.015). In the context of the POCR, the hazard ratio for the starting position factor marginally exceeded that of the breath protocol factor (0.753 vs. 0.667). Nonetheless, regarding the POCR, the Z-test outcome related to the breath protocol factor demonstrated statistical significance (*p* = 0.010). These findings were corroborated within the outcomes of the RSF analysis, as illustrated in Figure 5. According to Figure 5(1), in the case of the PHR, the starting position factor’s average absolute value of survSHAP(t) surpasses that of the breath protocol factor, signifying that the variable importance of the starting position factor exceeds that of the breath protocol factor. Similarly, as shown in Figure 5(2), for the POCR, the average absolute value of survSHAP(t) for the breath protocol factor is higher than that for the starting position factor, indicating that the variable importance of the breath protocol factor is greater than that of the starting position factor.

## 4. Discussion

The objective of this study is to ascertain whether adjustments in the initiation posture and the adoption of an appropriate breathing strategy can effectively mitigate the cardiopulmonary stress (PHR and POCR) induced by HIMEs (burpee exercise). There are three findings according to the results. First, the breath protocol significantly affects exercise performance. In the study results, under the condition of regulated breathing rhythm, participants completed noticeably more repetitions, and this was not affected by the choice of starting position. However, under the condition of free breathing rhythm, regardless of whether the starting position was stand or prone, the impact on overall repetitions was very limited. Second, under a regulated breathing rhythm condition, initiating the exercise in a prone position can mitigate cardiopulmonary stress, evidenced by a reduced PHR following burpee exercises commencing from a prone versus a standing starting position. Finally, beginning in a prone position appears to exert minimal influence on metabolic stress following burpee exercises and does not significantly enhance cardiopulmonary stress within the free breathing rhythm condition, aligned with the study’s exercise performance outcomes (namely, overall repetitions). It can be inferred that, on one hand, post-exercise cardiopulmonary and metabolic stress are primarily affected by exercise volume. On the other hand, changing the starting position can only affect post-exercise cardiopulmonary stress, and this can only be achieved when the exercise volume is similar.

This study found that the rhythm of breathing during exercise significantly affects performance. During testing, when participants strictly followed a regulated breathing rhythm, they were able to complete more repetitions in the burpee exercise at every moment of movement. As a fundamental mechanism of physiological regulation, breathing significantly influences athletic performance.

Research underscores that maintaining a consistent breathing rhythm enhances oxygen utilization efficiency, thereby optimizing blood and muscle oxygenation levels and boosting athletic performance [35,36]. A steady breathing pattern helps stabilize heart rate variability, which is crucial for the cardiovascular system’s regulation. It improves muscle blood circulation, extends endurance, and reduces the perception of fatigue [37]. Furthermore, a regulated breathing rhythm could potentially improve focus and decrease mental stress by affecting the central nervous system, thus enhancing overall exercise efficiency. Adapting specific breathing rhythm guidelines to match the biomechanical characteristics of different training activities within a regimen can significantly improve athletes’ performance and outcomes, heightening the efficacy of training programs [37,38]. Additionally, changing the breathing rhythm may have profound long-term effects on cardiopulmonary adaptation. For example, consistent deep breathing increases pulmonary ventilation and cardiac pump efficiency, which reduces the cardiac burden during exercise [39,40]. This enables a quicker return to resting heart rates post-exercise, thereby accelerating recovery from high-intensity activities. The optimization of cardiopulmonary function not only enhances immediate exercise performance but may also positively influence the long-term health and recovery capabilities of athletes [41]. These insights not only deepen the understanding of the role of regulated breathing rhythms in exercise but also provide vital strategic directions for future training plans and health management of athletes. It underscores the importance of further exploring the interactions between breathing rhythms and cardiopulmonary as well as metabolic functions, which will be a critical area of focus in sports science research going forward [42].

Nonetheless, this study found that without controlling the breathing rhythm, simply altering movement techniques is not sufficient to produce a significant positive impact on athletic performance. This finding emphasizes the irreplaceability of regular breathing in improving exercise efficiency [43]. The underlying mechanism may be related to the body’s demand for oxygen and energy during exercise [44]. Irregular breathing can result in an imbalance between oxygen intake and carbon dioxide expulsion, thereby impairing blood oxygenation efficiency and energy generation, rendering technical advancements less effective [45]. This finding has significant practical value, suggesting that individuals involved in developing and implementing training plans should not only focus on technical improvements but should also pay attention to the regularity of breathing rhythm [46].

Additionally, the findings of this study highlight that volume is a principal determinant impacting post-exercise cardiopulmonary and metabolic stress in high-intensity metabolic exercise (HIME). This finding implies that although trainers can complete a higher exercise volume by emphasizing breathing rhythm, it also leads to greater cardiopulmonary stress after training [47]. On one hand, a higher exercise volume can bring about greater training adaptation, but on the other hand, it also increases the risk of overtraining [48]. Consequently, prudence is essential in practical applications. The design of training protocols necessitates a balance between training volume and the body’s tolerance. When formulating training plans, adjustments to exercise volume should be made based on the individual’s physical status and recuperative capacity to circumvent physical injury and cardiopulmonary dysfunction stemming from overtraining [48]. Furthermore, a comprehensive training plan, including recovery strategies, should also be emphasized. For example, sufficient rest, nutritional supplementation, the control of sleep quality, and active recovery strategies (such as light stretching, massage, etc.) can help speed up recovery and reduce post-training cardiopulmonary system stress [49,50].

Lastly, another significant finding of this study is that optimizing the skill of training movements can have a positive impact on post-exercise cardiopulmonary and metabolic stress, but this is only meaningful when the exercise volume is similar. This implies that with a consistent exercise volume, enhancing movement skill can efficaciously boost exercise efficiency and curtail unnecessary energy expenditure and biomechanical burden, consequently mitigating strain on the musculoskeletal, cardiopulmonary, and metabolic systems [51,52]. The underlying rationale for this beneficial impact might be that heightened efficiency in movement skill lessens the body’s oxygen demand and carbon dioxide output, simultaneously diminishing undue stress on muscles and joints, rendering the exercise more economically viable, and ultimately easing the load on the cardiopulmonary and metabolic systems [53]. This finding emphasizes that in designing and implementing HIME training plans, in addition to reasonably controlling exercise volume, optimizing movement skill should also be prioritized.

This study has certain limitations. First, the absence of standardized breathing patterns among participants, who are general fitness enthusiasts rather than professional athletes, introduces variability that could influence the outcomes. Additionally, this study’s results may be affected by individual differences in habitual breathing techniques, which were not controlled or standardized. The exercise used in this study was the ‘Burpee‘, which does not include a pulling component. This may have introduced some bias in the results, potentially affecting the applicability of the findings to exercises like climbing or rowing, which involve pulling movements. Last but not the least, because of the strict method used to monitor the breathing pattern during the tests, if a participant failed, the test was stopped, and they had to rest before trying again. This caused some participants to need several attempts to complete the test. For these participants, the test volume increased, possibly leading to training adaptation and learning effects, which could result in their aerobic capacity being overestimated.

## 5. Conclusions

A regulated breathing rhythm might enhance performance significantly in HIMEs by allowing for increased exercise volume and optimizing physiological functions, including oxygen utilization and cardiovascular regulation. Conversely, changing the starting position without a focus on breathing rhythm might not markedly improve performance or reduce cardiopulmonary stress. The findings highlight the balance between exercise volume and skill optimization in mitigating post-exercise stress, suggesting that effective training protocol should prioritize both to maximize performance and recovery while minimizing the risk of overtraining and injury.

## Figures and Tables

**Figure 1 healthcare-12-01889-f001:**
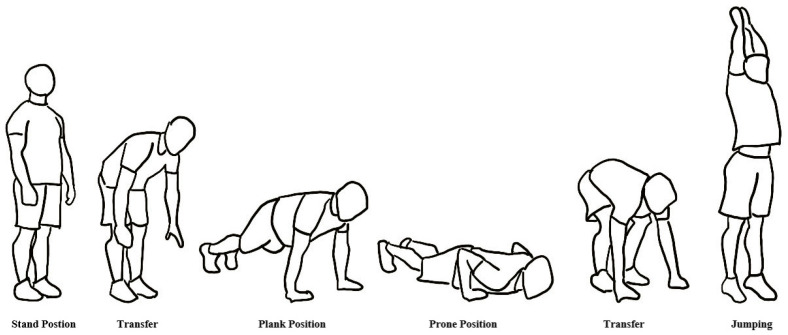
Process and positions of burpee movement.

**Figure 2 healthcare-12-01889-f002:**
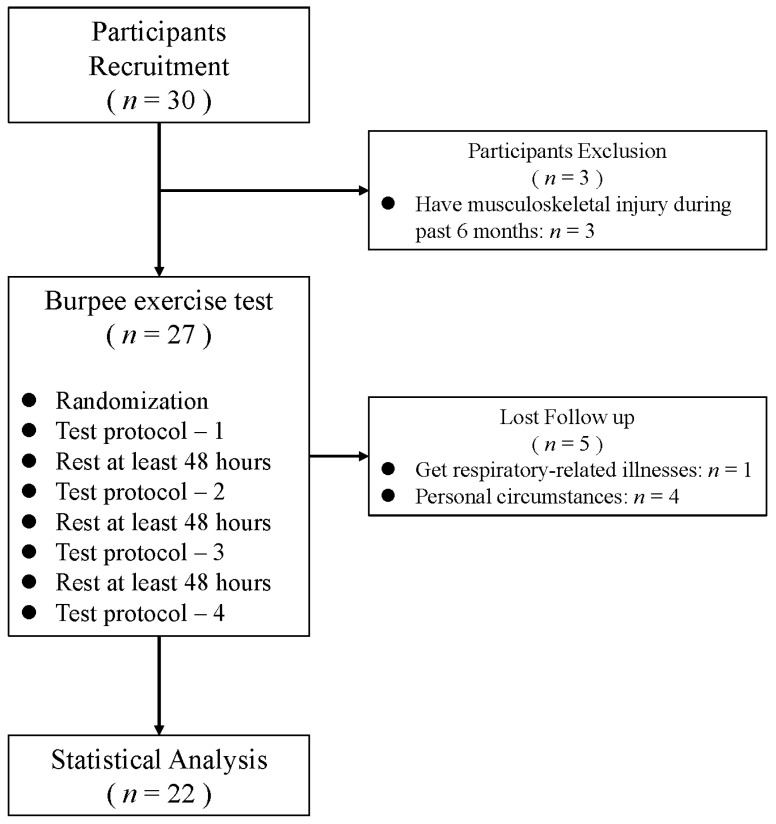
A flow diagram of the recruitment and testing process.

**Figure 3 healthcare-12-01889-f003:**
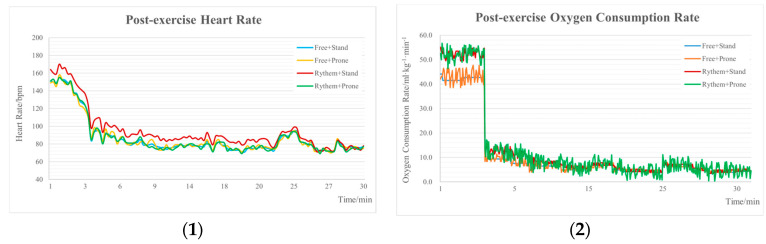
Trend of post-exercise heart rate and post-exercise oxygen consumption rate after different burpee exercise protocols. (**1**) Post-exercise heart rate; (**2**) post-exercise oxygen consumption rate.

**Figure 4 healthcare-12-01889-f004:**
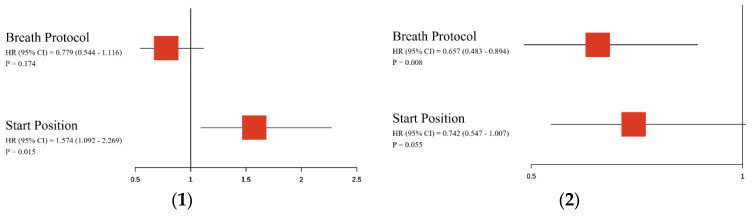
The forest plots of the multi-variable Cox-regression modeling of the PHR and POCR. (**1**) The PHR; (**2**) the POCR.

**Figure 5 healthcare-12-01889-f005:**
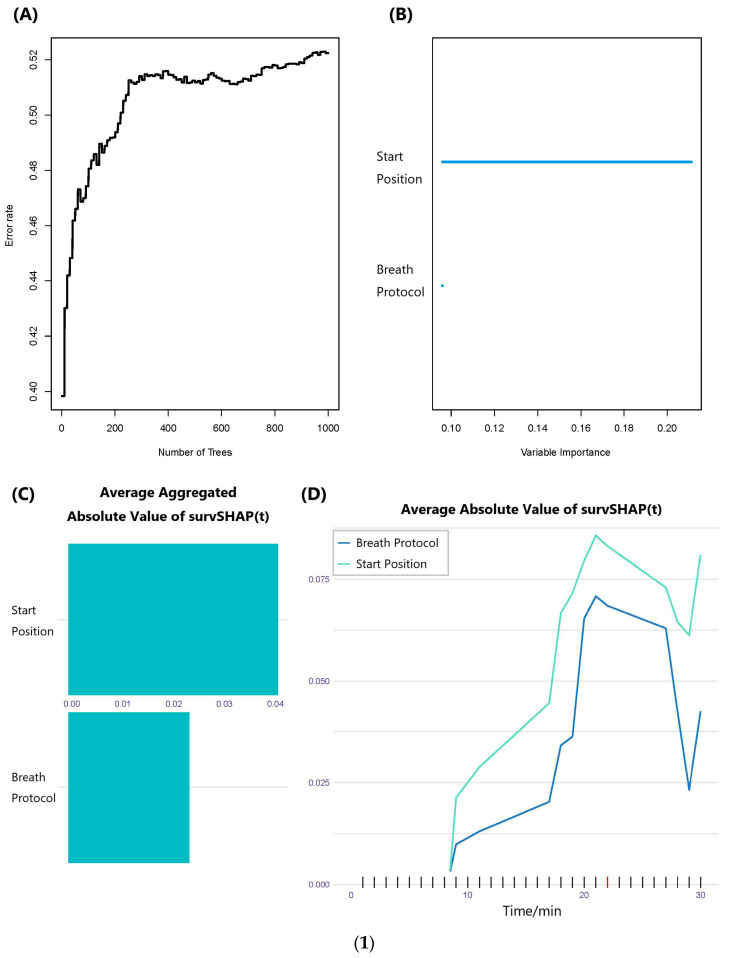
The results of the random survival forest analysis for the PHR and POCR. (**1**) The PHR; (**2**) the POCR. Note: Subplot (**A**) showed the error rate change within the number of trees created, subplot (**B**) was the bar plot of the variable importance, and subplots (**C**,**D**) showed the value and the trend of the average absolute value of survSHAP(t) over time post-exercise.

**Table 1 healthcare-12-01889-t001:** Requirements of breathing, positions, and conditions for stop test in each group.

Group	Description of Breathing and Positions			Conditions for Stop
	Starting Position	End Position	Time of Breathing	
RS	Standing Position	Standing Position	The participant was allowed to have 1-time breathing only between each two burpee movements (in standing position or prone position)	Cannot finish 20 burpees in 1 min OR Breathing when the participant was not allowed to.
RP	Prone Position	Prone Position		
FS	Standing Position	Standing Position	Anytime	Cannot finish 20 burpees in 1 min
FP	Prone Position	Prone Position		

RS: Regulated breathing rhythm + start from standing position; RP: regulated breathing rhythm + start from prone position; FS: free breathing rhythm + start from standing position; FP: free breathing rhythm + start from prone position.

**Table 2 healthcare-12-01889-t002:** The information of the eligible participants.

Items	Description		
**Sex (Male/Female)**	18/4		
**Age (Year)**	19.6 ± 1.2		
**Height (cm)**	168.4 ± 4.1		
**Body weight (kg)**	57.6 ± 5.2		
**Training experience (Year)**	3.3 ± 0.7		
**Maximum burpee in 1 min**	26.8 ± 1.2		
**Heart rate before test (bpm)**	**RS**	59.3 ± 2.9	*p* > 0.05
	**RP**	60.6 ± 2.5	
	**FS**	60.4 ± 2.9	
	**FP**	62.2 ± 2.6	
**Oxygen consumption before test (mL/kg/min)**	**RS**	36.4 ± 1.4	*p* > 0.05
	**RP**	36.5 ± 1.0	
	**FS**	37.0 ± 1.6	
	**FP**	37.0 ± 1.8	

RS: Regulated breathing rhythm + start from standing position; RP: regulated breathing rhythm + start from prone position; FS: free breathing rhythm + start from standing position; FP: free breathing rhythm + start from prone position.

**Table 3 healthcare-12-01889-t003:** Descriptive statistical analysis in different burpee exercise protocols.

Outcomes	Group	Mean (SD)	Two-Way ANOVA		Post hoc Test with Bonferroni–Holm Correction			
			F	*p*-Value	Mean Difference			
**Overall repetitions**	FS	55.7 (2.44)	761.62	<0.001	FS	−1.18 (0.60)	−20.73 (0.60) *	−20.68 (0.60) *
	FP	56.9 (1.42)				FP	−19.55 (0.60) *	−19.50 (0.60) *
	RS	76.4 (1.79)					RS	0.05 (0.60)
	RP	76.4 (2.11)						RP
**Maximum** **HR in test** **(bpm)**	FS	161 (4.41)	41.2	<0.001	FS	3.09 (1.33)	−9.32 (1.33) *	3.82 (1.33)
	FP	158 (5.56)				FP	−12.41 (1.33) *	0.72 (1.33)
	RS	170 (4.01)					RS	13.14 (1.33) *
	RP	157 (3.41)						RP
**Average** **HR in test** **(bpm)**	FS	84.4 (4.82)	35.39	<0.001	FS	−1.68 (1.10)	−10.55 (1.10) *	−4.09 (1.10)
	FP	86.0 (3.03)				FP	−8.86 (1.10) *	−2.41 (1.10)
	RS	94.9 (3.80)					RS	5.81 (1.10) *
	RP	94.7 (1.86)						RP
**Maximum** **PHR** **(bpm)**	FS	160 (4.02)	39.7	<0.001	FS	2.65 (1.39)	−9.95 (1.39) *	3.45 (1.39)
	FP	157 (5.78)				FP	−12.6 (1.39) *	0.80 (1.39)
	RS	170 (3.95))					RS	13.40 (1.39) *
	RP	157 (3.41)						RP
**Average** **PHR** **(bpm)**	FS	83.9 (4.80)	31.72	<0.001	FS	−2.10 (1.16)	−10.65 (1.16) *	−4.55 (1.16) *
	FP	86.0 (3.15)				FP	−8.55 (1.16) *	−2.41 (1.16)
	RS	94.5 (3.72)					RS	6.45 (1.16) *
	RP	88.5 (2.61)						RP
**Maximum** **POCR** **(mL/kg/min)**	FS	45.5 (2.02)	160.87	<0.001	FS	−3.15 (0.70) *	−11.74 (0.70) *	−12.60 (0.70) *
	FP	48.7 (2.37)				FP	−8.59 (0.70) *	−9.45 (0.70) *
	RS	57.3 (2.17)					RS	−0.86 (0.70)
	RP	58.1 (2.65)						RP
**Average** **POCR** **(mL/kg/min)**	FS	11.0 (0.93)	45.85	<0.001	FS	−0.58 (0.35)	−3.16 (0.35) *	−3.17 (0.35) *
	FP	11.7 (0.98)				FP	−2.59 (0.35) *	−2.60 (0.35) *
	RS	14.3 (1.28)					RS	0.01 (0.35)
	RP	14.2 (1.48)						RP

*: statistically significant, *p* < 0.008; FS: free breathing rhythm and standing position; FP: free breathing rhythm and prone position; RS: regulated breathing rhythm and standing position; RP: regulated breathing rhythm and prone position; PHR: post-exercise heart rate; POCR: post-exercise oxygen consumption rate; SD: standard deviation.

**Table 4 healthcare-12-01889-t004:** The multi-variable Cox-regression model of the PHR and POCR over time.

Outcome	Factors	B	SE	HR (95% CI)	Z Test	
					Z	*p*-Value
**PHR**	Breath Protocol	−0.249	0.183	0.779 (0.544, 1.116)	−1.36	0.174
	Start Position	0.454	0.187	1.574 (1.092, 2.269)	2.434	0.015
**POCR**	Breath Protocol	−0.405	0.157	0.667 (0.491, 0.906)	−2.59	0.010
	Start Position	−0.284	0.155	0.753 (0.555, 1.020)	−1.835	0.067

B: regression coefficient; SE: standard error; HR: hazard ratio; PHR: post-exercise heart rate; POCR: post-exercise oxygen consumption rate.

## Data Availability

Data are contained within the article.

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
