# Peer review of "The Impact of Starting Positions and Breathing Rhythms on Cardiopulmonary Stress and Post-Exercise Oxygen Consumption after High-Intensity Metabolic Training: A Randomized Crossover Prospective Study"

_healthcare, 2024, doi:10.3390/healthcare12181889_

Round 1
Reviewer 1 Report
Comments and Suggestions for Authors
The topic of the study “Impact of Starting Positions and Breathing Rhythms on Cardiopulmonary Stress and Post-Exercise Oxygen Consumption after High-Intensity Metabolic Training: A Randomized Crossover Prospective Study” is interesting, however, I think there are some issues that should be considered.
INTRODUCTION
The introduction does not talk about how breathing rhythm could improve recovery. The mechanisms by which this improvement occurs (if there can be one).
METHODS
I have doubts about how it has been controlled that each participant has applied the breathing pattern correctly. That is, it is one thing to indicate that you should inhale or exhale at a specific moment and another to ensure that this has been done since it is something that cannot be perceived with the naked eye. On the other hand, it would be necessary to explain the reason for each pattern in each position.
In relation to the tests, it is clear that the order was randomized, but have the baseline differences between the same athlete been analyzed? That is, the perception of fatigue, basal heart rate, respiratory rate, etc. are variables highly conditioned by the context. It is important to know if each test has been done under the same physical and mental conditions.
Another question would be to know the respiratory pattern of each sportist. Has this been asked? How can a change in pattern influence post-exercise stress? If the athlete has managed to be efficient with a specific pattern, can't changing it generate more stress?
RESULTS
I think the results are well presented but they are incomplete in relation to the points I have raised in the methodology section. Although the representation of burpees is not clear to me. If all participants had to complete 20 burpees per minute...if they have not done so, has the result been considered? If it is like that from the start there would be differences in physical condition. Has this been evaluated?
DISCUSSION
The same goes for discussion. I think there is a lack of reasoning to support the results, especially thinking about the more physiological part.
Reviewer 2 Report
Comments and Suggestions for Authors
-Lines 73-74 are not complete.
-Lines 76-77 need to have the English improved.
Round 2
Reviewer 1 Report
Comments and Suggestions for Authors
Dear authors, congratulations on the effort to improve the manuscript, it is now much clearer. However, the limitations of the work are great, especially one that I already mentioned in the first review. I think it is difficult to know whether or not athletes have complied with the breathing pattern they should apply at all times. It has not been recorded in any way and it is not possible to know if this breathing pattern has been maintained throughout each test or if each athlete has returned to their natural pattern when fatigue has appeared. This directly influences the results and conclusions of the work since it would completely change the reasoning.
Author Response
Comments: Dear authors, congratulations on the effort to improve the manuscript, it is now much clearer. However, the limitations of the work are great, especially one that I already mentioned in the first review. I think it is difficult to know whether or not athletes have complied with the breathing pattern they should apply at all times. It has not been recorded in any way and it is not possible to know if this breathing pattern has been maintained throughout each test or if each athlete has returned to their natural pattern when fatigue has appeared. This directly influences the results and conclusions of the work since it would completely change the reasoning.
Response: Thank you for your valuable feedback and for your continued support in enhancing the clarity and quality of our work. We greatly appreciate your insights and the time you have taken to review our manuscript. We acknowledge the importance of controlling the breathing pattern during the tests, as it directly impacts the validity of our results and conclusions. To address this concern, we have re-conducted the whole test and replaced the previous K3 version of the FitMate® EMD gas metabolism analyzer with the K5 version, which wis wireless and wearable (Page 4, Lines 199 to 204). The K5 version enables continuous monitoring of the ventilatory parameters, allowing us to more effectively control the breathing pattern throughout the tests. This upgrade ensures that any deviations in breathing patterns are promptly detected, and appropriate actions are taken to maintain the integrity of the data. To provide a clear understanding of the steps we have taken, we have revised the manuscript on Page 4, Lines 204 to 210 to include a detailed description of the method used to monitor the breathing pattern.
"During the test, a researcher continuously monitored the breathing pattern of the par-ticipants by observing the ventilatory parameters displayed on the screen of the gas metabolism analyzer. If a participant took a breath outside the designated time of breathing, the screen would immediately show a change in ventilatory values. In such cases, the test would be deemed a failure and promptly terminated, and the participant would be asked to have a full rest. The test would rescheduled and conducted again after a 48-hour rest period."
Moreover, we have also considered the implications of implementing this more stringent method for monitoring the breathing pattern. As a result, some participants required multiple attempts to successfully complete a test, which increased the overall test volume. This may have introduced potential training adaptation and learning effects in the participants. We have added this consideration to the limitation statement in the discussion section on Page 13, Lines 497 to 501.
“Last but not the least, because of the strict method used to monitor the breathing pattern during the tests, if a participant failed, the test was stopped, and they had to rest before trying again. This caused some participants to need several attempts to complete the test. For these participants, the test volume increased, possibly leading to training adaptation and learning effects, which could result in their aerobic capacity being overestimated.”
We believe these revisions address the concerns raised and enhance the robustness of our study. We hope the revised manuscript meets your expectations, and we look forward to your feedback. Thank you once again for your constructive comments.
Reviewer 2 Report
Comments and Suggestions for Authors
Nice job on the edits.
Author Response
Comments: Nice job on the edits.
Response: Thank you very much for your help to our work!